# Association between Depression, Anxiety Symptoms and Gut Microbiota in Chinese Elderly with Functional Constipation

**DOI:** 10.3390/nu14235013

**Published:** 2022-11-25

**Authors:** Jiajing Liang, Yueming Zhao, Yue Xi, Caihong Xiang, Cuiting Yong, Jiaqi Huo, Hanshuang Zou, Yanmei Hou, Yunfeng Pan, Minchan Wu, Qingqing Xie, Qian Lin

**Affiliations:** 1Department of Nutrition Science and Food Hygiene, Xiangya School of Public Health, Central South University, 110 Xiangya Rd, Changsha 410078, China; 2Hyproca Nutrition Co., Ltd., Changsha 410011, China; 3Department of Epidemiology, School of Public Health, Sun Yat-sen University, Guangzhou 510275, China

**Keywords:** depression, anxiety, constipation, intestinal flora, elderly

## Abstract

This study aimed to investigate the relationship between anxiety, depression, and gut microbiota in elderly patients with FC. Methods: in this cross-sectional study, a total of 198 elderly participants (85 male and 113 female) aged over 60 years were recruited. The study was conducted in Changsha city, China. The participants completed an online questionnaire, including The Patient Health Questionnaire-9 (PHQ-9), Generalized Anxiety Disorder-7 (GAD-7), The Patient Assessment of Symptoms (PAC-SYM), and The Patient Assessment of Quality of Life (PAC-QoL). We selected the 16S rDNA V3 + V4 region as the amplification region and sequenced the gut microbiota using the Illumina Novaseq PE250 high-throughput sequencing platform. Results: in total, 30.3% of patients with constipation had depression, while 21.3% had anxiety. The relative abundance of intestinal microbiota in the normal group was higher than that in the anxiety and depression group. According to LEfSe analysis, the relative abundance of *g_Peptoniphilus* and *g_Geobacter* in the people without depression and anxiety was higher. The relative abundance of *g_Pseudoramibacter-Eubacterium* and *g_Candidatus-Solibacter* in the depression group was lower, and the relative abundance of *g_Bacteroides* and *g_Paraprevotella, g_Cc_115* in the anxiety group was higher. In addition, according to the correlation analysis, *g_Aquicella* and *g_Limnohabitans* were negatively correlated with constipation symptoms, anxiety, and depression. Conclusions: this study found that gut microbiota composition may be associated with a higher incidence of anxiety and depression in patients with FC, thus providing insight into the mechanisms that ameliorate mood disorders in patients with FC.

## 1. Introduction

Functional constipation, with a prevalence of 15% [1] worldwide, is a functional gastrointestinal disorder with no organic lesions or physiological abnormalities. Individuals with persistent constipation tend to self-medicate and abuse drugs, which could affect the optimal timing of treatment and exacerbate the disease. As a result, this may impact quality of life and lead to other chronic diseases [2]. Moreover, a set of psychological variables may be associated with constipation; several studies revealed anxiety and depression as potential mediators of the relationship between constipation severity and quality of life [3,4,5,6]. Constipation is common among people over 60 years old [7] and, in China, about 22% of the elderly suffer from this condition [8]. Immobility, malnutrition, rectal insensitivity, pelvic floor muscle weakness, chronic disease, and long-term medication increase the risk of constipation and psychiatric problems (anxiety or depression) in the elderly population [9,10]. It has been found that psychiatric problems are closely related to constipation risk, especially in the elderly [11]; however, overall, there is still little research on this issue. Examining the relationship between functional constipation and mental disorders among the elderly population may provide evidence that will help in improving quality of life in the aging population.

The gut microbiome, which refers to the microbial community in the human gastrointestinal tract, is a potential common cause of digestive tract diseases and mental disorders. An increasing number of studies demonstrate that the occurrence of FC is closely associated with the imbalance of gut microbiota [12,13]. Among the FC patients, the abundance of Bacteroides, Rosiella, and Faecococcus in the intestine was decreased, while pathogenic bacteria rich in genes related to the methanogenic pathway, hydrogen production, and glycerol, such as Escherichia coli and fungi, were significantly increased [14,15]. Additionally, the “brain–gut axis” plays a bidirectional role in the relationship between mental disorders and gut microecology. One study argued that the abundance of Clostridium and actinomycetes in the intestinal tract of depressed people was higher than that in healthy people [16]. Another study [17] found that depressed people had lower levels of Bacteroidetes and Lactobacillus than healthy people. A fecal transplant intervention trial revealed that, when gut microbiota diversity was increased, depression and anxiety symptoms improved in patients with FC [18]. Several studies focused on the association between FC and mental disorders, whereas few studies have analyzed the role of gut microbiota in this interaction.

Therefore, this study investigated the relationship between intestinal microbiota, and depression and anxiety symptoms among the elderly with FC in China. We hypothesized that FC is associated with anxiety and depression, in which the gut microbiota play a specific role.

## 2. Materials and Methods

### 2.1. Ethic Approvals

The study was approved by the Ethics Review Committee of the Xiangya School of Public Health, Central South University (XYGW-2021-54).

### 2.2. Study Design and Participants

This was a cross-sectional study. From August 2021 to January 2022, we conducted this study in the Community Health Service Center of Wangluyuan Sub-district in Changsha, the capital city of Hunan Province in South Central China. According to the inclusion and exclusion criteria, elderly participants (1) age ≥ 60 years were recruited. (2) In accordance with the FC diagnostic criteria of Rome IV [19], IBS and opioid-induced constipation were excluded, and participants meeting two or more of the following criteria were included: straining in more than 25% of defecations; lumpy or hard stools in more than 25% of defecations; sensation of incomplete evacuation in more than one-quarter (25%) of defecations; sensation of anorectal obstruction/blockage in more than one-quarter (25%) of defecations; manual maneuvers to facilitate more than one-fourth (25%) of defecations; fewer than three spontaneous bowel movements per week. (3) Participants had no physical disability, serious cardiovascular disease, serious gastrointestinal disease, serious liver or kidney disease, or endocrine disease, and (4) were sufficiently literate and able to understand the study. The exclusion criteria were: living in Changsha, Hunan Province, China for less than 6 months; using probiotics, dietary fiber, or antibiotics in the last 3 months; surgery in the last 6 months; suspected organic lesions, accompanied by fever, bloody stool, or melena. A total of 198 elderly people completed the questionnaire and submitted stool samples.

### 2.3. Measures

The survey included a questionnaire, anthropometric measurements, and fecal intestinal flora detection.

#### 2.3.1. Questionnaire Survey

Elderly participants were interviewed face to face by a uniformly-trained research assistant in order to complete the questionnaire. The Questionnaire Star, an online questionnaire tool, was used to develop and deliver the investigation questionnaire. Each questionnaire had a unique linkage. We collected demographic information about older adults, constipation-related conditions, mood status, and other influencing factors.

Demographic information: age, gender, educational background, occupation, marital status, family monthly income, etc.;Constipation-related symptoms: the Patient Assessment of Symptoms (PAC-SYM) was used to assess the symptoms of constipation in the study subjects [20]. The PAC-SYM consists of 12 items in 3 dimensions (abdominal symptoms, rectal symptoms, and stool symptoms). The Likert 5-level scoring method was used to assign 0–4 points to “no such symptoms”, “mild”, “moderate”, “severe”, and “very severe”, respectively. The score of each dimension was the average score of all items in this dimension, and the total score was the average score of all items. The higher the score, the more severe the constipation symptoms. This study investigated the symptoms of constipation in participants over the previous week. In this study, the total scores of all the subjects were divided into three grades according to the percentile, with mild symptoms lower than 33.3%, severe symptoms higher than 66.7%, and moderate symptoms in the rest;Constipation-related quality of life: the Patient Assessment of Quality of Life (PAC-QOL) was used to assess the impact of constipation on quality of life over the previous two weeks [21]. This scale consists of 28 items divided into 4 facets, including physical discomfort, psychosocial discomfort, anxiety, and satisfaction. The score ranges from 0 to 112, with a lower score indicating a higher quality of life. In this study, the total scores of all the subjects were divided into three grades according to the percentile. A score lower than 33.3% indicated high quality of life, a score higher than 66.7% indicated low quality of life, and the remaining scores indicated medium quality of life;Depressive symptoms: Patient Health Questionnaire-9 (PHQ-9) was used to assess whether older people had experienced depressive symptoms in the past two weeks [22,23]. The PHQ-9 consists of nine items on a four-point response scale with scores ranging from “0” to “3” for “None”, “Few days”, “More than half of the days”, and “Almost every day”, respectively. The overall score ranges from 0 to 27, with a higher score meaning a higher degree of depressive symptoms. Participants with a score of 5 or higher were defined as “at least mild depression” [22,23].;Anxiety symptoms: Generalized Anxiety Disorder-7 (GAD-7) was used to investigate the anxiety status of the participants. GAD-7 consists of seven items on a four-point response scale (from 0, never, to 3, almost every day). Participants with a score of 5 or higher were defined as “at least mild anxiety” [22,23];Dietary intake: the Food Frequency Questionnaire (FFQ) was used to investigate the dietary intake of elderly patients with FC over the previous month. According to “Chinese Food Composition Table (Standard Edition, Volume I and Volume II)” [24], foods with fiber higher than 3.0/100 mg/g [25] were selected as the items of FFQ. The consumption frequency of each food item was asked and the response format ranged from “never” to “three times a day”. Then, the intake frequency of 37 kinds of food was recoded into times per week: never eaten = 0, <1 time/month = 0, 1~3 times/month = 0.5 times/week, 1~2 times/week = 1.5 times/week, 3~4 times/week = 3.5 times/week, and 5~6 times/week = 5.5 times/week; 1 times/day = 7 times/week, 2 times/day = 14 times/week, and 3 times/day or more = 21 times/week or more. In this study, the frequency of high dietary fiber intake in all subjects was divided into three grades according to tertile. The frequency of low fiber intake was lower than 33.3%, high fiber intake was higher than 66.7%, and the remaining participants had a medium intake;Physical activity level: physical activity was assessed using the short-form self-administered instruments of the International Physical Activity Questionnaire (IPAQ) [26]. The questionnaire consisted of seven items and used indicators in MET min/week (Metabolic equivalence, MET) to indicate the intensity of physical activity. The IPAQ was scored according to recommended guidelines [27]. The physical activities were categorized into low, moderate, and high, based on IPAQ guidelines. Physical activity levels were divided into three grades: low activity or inactivity, moderate physical activity, and high physical activity;Sleep Quality: Pittsburgh Sleep Quality Index (PSQI) was used to evaluate the Sleep Quality of the elderly in the past 1 month. Eighteen self-assessment items composed of seven components were involved in the scoring, and each component was scored according to the 0~3 level. The cumulative score of each component was the total PSQI score. The total score ranges from 0 to 21, with higher scores indicating worse sleep quality. A score of 5 or less indicates good sleep quality, while a score of more than 5 indicates poor sleep quality [28].

#### 2.3.2. Anthropometry

Conducted by uniformly-trained research assistants, the height was measured using a standard height meter and weight with a TANITA human body composition analyzer BCW02C (Guangdong Food and Drug Administration (prospective), No. 2210704, 2014). Body mass index (BMI) was calculated by dividing body weight (kg) by height squared (m^2^). According to the recommendations of the Dietary Guidelines for Chinese Residents (2022) [29], the normal range of BMI in the elderly in this study is 20.0–26.9, lower than the normal range indicates emaciation and 27.0 and above indicates obesity.

#### 2.3.3. Fecal Collection

The fecal microbial genome protection solution set provided by Wekemo Tech Group Co., Ltd., (Shenzhen, China) was used in this study, and could be stored at room temperature for 7 days. On the day of the questionnaire survey, after the sampling method and precautions were explained by the research team members, feces were collected by the respondents at home and sent to the community health service center within 3 days. The feces were brought back to the laboratory by the research team members and stored in an ultra-low-temperature refrigerator at −80 °C. All the samples were collected and sent to Wekemo Tech Group Co., Ltd. For sequencing analysis. The 16S rDNA V3 + V4 region was selected as the amplified region and sequenced by the Illumina Novaseq PE250 high-throughput sequencing platform.

### 2.4. Data Analysis

#### 2.4.1. Statistical Analysis

The IBM SPSS 24.0 software (IBM Corp., Armonk, NY, USA) was used for data entry and analyses. Descriptive data were presented as a percentage or the mean with standard deviation. We used a Chi-squared test to analyze related variates for different groups, and Spearman’s rank correlation coefficient was used to analyze the association between emotional status and FC symptoms in the elderly. In all the analyses, *p* < 0.05 was considered statistically significant.

#### 2.4.2. Analysis of Intestinal Flora

The analysis was conducted by following the “Atacama soil microbiome tutorial” of Qiime2docs along with customized program scripts (https://docs.qiime2.org/2022.8/, accessed on 22 November 2022). The Qiime2 diversity plug-in was used to analyze alpha diversity and beta diversity. Alpha diversity index is the analysis of species diversity in a sample, including the richness and evenness of species composition in the sample. The Shannon, Chao-1, and other indexes are usually used to evaluate the species diversity of a sample. The higher the index, the more complex the diversity of the sample. The Shannon index is calculated considering the total operational taxonomic units (OTUs) in the sample and the proportion of each € [30]. The Chao1 index is a measure of species richness (the number of species). It extrapolates a theoretical richness from the observed results that is closer to the true richness. After obtaining the overall alpha diversity index, the Kruskal–Wallis method was used to determine whether there was a significant difference in alpha diversity index between each sample group when combining the grouping information.

Beta diversity is the comparison of microbial community composition among different samples. PCoA analysis was performed based on Bray–Curtis distance, weighted Unifrac distance, and unweighted Unifrac distance, and the principal coordinate combination with the largest contribution rate was selected for the mapping display. The closer the distance between samples in the figure, the more similar the species composition structure of the samples. After obtaining the overall beta diversity index, the PERMANOVA method and ANOSIM method were used to compare whether there were significant differences in microbial composition and structure between each sample group [31].

We used the linear discriminant analysis of effect size (Lefse), an analytical tool for discovering and interpreting biomarkers in high-dimensional data, to find microbiota with a high abundance in different groups [32]. In this study, the threshold of LDA was 2.

The Pheatmap package in R language was used to draw a correlation heatmap, and the Spearman correlation coefficient was calculated to analyze whether environmental factors (three dimensions of constipation symptoms, PHQ-9, and GAD-7) were significantly correlated with microbial communities or species.

## 3. Results

### 3.1. General Characteristics of Participants

A total of 198 eligible elderly people participated in this study. The average age was 72.1 ± 6.6 years old (Table 1), and the average BMI was 24.2 ± 3.3 kg/m^2^ (Table 1). More than half (57.1%) of the study participants were women, and there were no differences in the levels of constipation symptoms between males and females (*p* = 0.868). Among the participants, 83.8% had finished junior high school or above. The higher the PAC-SYM, the poorer the quality of life concerning constipation (*p* < 0.001). Constipation symptoms were more severe in older adults with poor sleep quality compared to those with better sleep quality (*p* = 0.003). Dietary fiber intake varied at constipation symptom levels (*p* = 0.004). An association between physical activity and constipation symptoms was not observed (*p* > 0.05).

### 3.2. Relationship between Depression, Anxiety and FC

In the present study, 30.8% and 21.7% of FC elderly suffered depression and anxiety, respectively (Table 2). The proportion of depression and anxiety increased with the aggravation of constipation symptoms (Depression accounted for Light: 17.5% vs. Moderate: 28.2% vs. Severe: 46.0%, *p* = 0.003; Anxiety accounted for Light:8.8% vs. Moderate: 21.8% vs. Severe: 33.3%, *p* = 0.005). Similarly, older adults with a high quality of life were less likely to be depressed or anxious (Depression accounted for Low: 54.5% vs. Moderate: 30.2% vs. High: 8.7%, *p* = 0.005; Anxiety accounted for Low: 43.9% vs. Moderate: 15.9% vs. High: 5.8%, *p* < 0.001).

We analyzed the association between the scores of each dimension of constipation symptoms and PHQ-9 and GAD-7 scores (Figure 1). The change in PHQ-9 score was positively correlated with the change in each dimension of constipation symptoms. The change in GAD-7 score was positively correlated with abdominal symptoms and the total score of constipation symptoms, but had no significant correlation with fecal traits and rectal symptoms. Spearman’s coefficient was shown in three dimensions, and abdominal symptoms were positively correlated with depression and anxiety (Depression: r = 0.28, *p* < 0. 001. Anxiety: r = 0.25, *p* < 0.001). In addition, rectal symptoms had the highest correlation coefficient with constipation symptoms (r = 0.91, *p* < 0.001).

### 3.3. Intestinal Microbiota Analysis Results

For depression, the Venn diagram shows that a total of 2253 OTUs are common between the two groups, and 3875 OTUs are specific to the depressed group (Figure 2a). The relative abundance composition of intestinal microbiota at the phylum level was similar between the depressed group and the non-depressed group (Figure 2b). There was no difference in the Shannon and Chao-1 indexes of alpha diversity in elderly participants with FC (Figure 2c,d). Figure 2e shows that there is a statistically significant difference in intestinal flora composition between the two groups (*p* = 0.034). This indicates that the composition of intestinal microbiota was different between the depressed and non-depressed groups.

For anxiety, the below Venn diagram shows that a total of 2034 OTUs are common between the two groups, and 2560 OTUs are specific to the anxiety group (Figure 3a). The relative abundance composition of intestinal microbiota at the phylum level is similar between the anxiety group and the non-anxiety group (Figure 3b). There is no difference in the Shannon and Chao-1 index of alpha diversity in the elderly with FC (Figure 3c,d). Figure 3e indicates that there is no statistical difference in the composition of intestinal microbiota between the two groups (*p* = 0.169).

LEfSe analysis showed that the difference in intestinal microbiota between the depressed and non-depressed groups demonstrates that, among the depressed participants, *s_Fragilis* and *s_Ovatus* had a higher abundance of microbiota. The abundances of *s_Copri, g_Pseudoramibacter-Eubacterium, g_Candidatus-Solibacter, g_Peptoniphilus,* and *g_Geobacter* were higher in the non-depressed population (Figure 4a). In the population with anxiety, the flora with higher abundances were *s_Fragilis, s_Producta, g_Bacteroides, g_Paraprevotella,* and *g_Cc_115*, and in the in the population with an absence of anxiety, they were *s_Perfringens, g_Granulicatella, g_Streptococcus, g_Peptoniphilus,* and *g_Geobacter* (Figure 4b).

According to the heat map (Figure 5), it can be seen that, at the genus level, rectal symptoms and abdominal symptoms, which are the two major facets of constipation (PAC-SYM), were mostly changed in the same direction with emotional problems. Fecal traits were positively associated with *g_Eubacterium, g_Ruminococcus,* and *g_Aggregatibacter*. Both mental disorders and abdominal symptoms were negatively associated with *g_Aquicella*. *G_Limnohabitans* was negatively associated with both mood disorders and rectal symptoms. The Class Actinobacteria were negatively associated with depression and anxiety.

## 4. Discussion

Emotional problems are common in individuals with FC [33]. This study aimed to explore the role of gut microbiota in the relationship between FC and mental health. We found that about one-third of elderly people with FC have depression or anxiety disorders. The abundance of Actinobacteria increased while *g_Aquicella* and *g_Limnohabitans* decreased among individuals without mental health conditions. Our results indicate that gut microbiota play a significant role in the development of constipation and mental health. Furthermore, this is a potential approach to improve both emotional disorders and intestinal diseases.

In this study, the prevalence of depression and anxiety disorders among the elderly was 30.3% and 21.3%, respectively, higher than those of general older adults in China (most of whom were aged 60 years or over), with a prevalence of depression of 23.6% and anxiety of 16.6% [34]. A possible explanation might be that insomnia, pain, or poor treatment resulting from constipation could seriously affect individuals’ quality of life, with elderly people particularly at risk [35]. Our results also confirm that, as constipation symptoms worsened, so did depression and anxiety emotions, which is consistent with findings from a large cross-sectional study [36]. In addition, some studies have shown that anxiety or depression factors in patients with FC, such as adrenal-corticosteroid-releasing factor and serotonin levels [37,38], may be related to the gut–brain axis pathway, in which the gut microbiota play an integral role [39,40].

Our results demonstrate that, among the FC elderly, those with depression/anxiety emotions had a lower out level than that of those without emotional disorders. However, there was no difference in alpha index between the groups with or without depression/anxiety, which is similar to the study of Dong Z et al. [41], but inconsistent with the study of Huang et al., which revealed that the α diversity of gut microbiota decreases in patients with mental diseases [42]. This contradictory result may be due to the differences in the severity of emotional problems. Regarding the β diversity, which is the index of microbial community compositions in different individuals, the depressed group had a more minor inter-individual difference than the non-depressed group. This is consistent with previous studies indicating that microbial β diversity is associated with depressive symptoms [43,44,45]. There was no difference in beta diversity between the two groups with or without anxiety, which is similar to the review results of Nikolova VL et al. [46]. The possible reason for this is that the anxiety problems were not very serious, and beta diversity will not change when the anxiety problem is mild [47].

The results from the LEfSe analysis show that the relative abundances of *g_Candidatus-Solibacter, g_Pseudoramibacter-Eubacterium, g_Peptoniphilus,* and *g_Geobacter* were lower in the FC elderly with depressive symptoms (PHQ-9 ≥ 5) than in those without depressive moods, and the abundance of *g_Candidatus-Solibacter* was negatively correlated with the PHQ-9 score, which agrees well with the study of Yu M et al. [48]. The reason for this might be that *g_Candidatus* is closely related to depression through purine metabolism and fatty acid metabolism [49]. This indicated that *g_Candidatus* might have the potential to improve both depression and constipation symptoms. Moreover, the abundance of *g_Peptoniphilus* and *g_Pseudoramibacter_Eubacterium* was also negatively associated with PHQ-9 scores, demonstrating that, the lower the proportion of these two bacteria in the intestine, the more serious the depression of FC individuals. Acetate, the major metabolite of *g_Peptoniphilus,* could regulate mood and create an acidic environment to promote defecation [50]. *g_Pseudoramibacter-Eubacterium,* the primary producer of short-chain fatty acids, can provide energy for intestinal cells and act as an anti-inflammatory factor in the intestinal tract by metabolizing dietary fiber [51]. However, studies [52] have shown that Eubacter has a high abundance in patients with constipation, which requires further study in the future.

The results of anxiety from LEfSe analysis indicate that FC individuals with higher GAD-7 scores had a lower relative abundance of *g_Granulicatella, g_Turicibacter, g_Peptococcus,* and *g_Peptostreptococcus,* and a higher abundance of *g_Bacteroides, g_Paraprevotella,* and *g_Cc_115*. In accordance with the results from Valles-Colomer M et al. [53], this may be because the abundance of *g_Bacteroides* is related to a high-calorie diet [54], which is usually accompanied by anxiety and depression. According to the study of Costa LM et al. [55], the abundance of *g_Bacteroides* was positively correlated with defecation effort because they were related to a proinflammatory diet. This suggests that *g_Bacteroides* may be involved in the severity of anxiety and constipation. In the current study, the relative abundance of *g_Actinobacteria, g_Aquicella,* and *g_Limnohabitans* in depression and anxiety groups was significantly lower than that of the opposite group. This study [56] shows that the alterations of OTU are associated with the short chain fatty acid (SCFA)-producing bacteria. Actinobacteria can promote the production of SCFA, which has a substantial impact on emotional health via the microbiota–gut–brain axis and through the Bifidobacteriaceae family [57]. Additionally, there are some floras associated with constipation symptoms. The abundance of *g_Bacteroides* and *g_Streptococcus* is positively associated with exertion in defecation and the hardness of the stool, respectively [55]. This correlation could be explained by the proinflammatory effects of the two bacteria [58,59]. However, similar to the study of Jun Xie et al., the abundance of *g_Streptococcus* and GAD-7 scores was inversely proportional [60], and the occurrence of this abnormality in *g_Streptococcus* may be due to functional specificity. Our results may provide new ideas for using intestinal flora to improve both FC and emotional disorders, and further research is urgently required.

Notably, our study analyzed the potential influencing factors of depression and anxiety, and we found that FC elderly with a lower BMI [61,62], poor sleep quality [63,64], and less frequent fiber consumption [65,66] had higher scores of PHQ-9 and GAD-7 (detailed results are shown in Appendix A of the Appendix A). These findings are in line with those of previous studies, reminding us that elderly people with FC must maintain an appropriate BMI, improve sleep quality, and increase the consumption of high-fiber foods and physical activity level [67,68] in order to prevent depression and anxiety.

This is the first study to focus on the relationship between the gut microbiota and mental health and FC in older adults. The main strengths of this study are its relatively large sample size and standard quality control. In addition, we considered confounding variables, including the high intake of dietary fiber and physical activities associated with constipation, which creates more comparable and reliable results. Moreover, all surveys were conducted face-to-face by uniformly-trained investigators to ensure the authenticity and completeness of the data.

However, some limitations in the interpretation of our results should be considered. First, due to the cross-sectional design of this study, the causal relationship between depression, anxiety and FC, and other influencing factors may not be confirmed. Second, we only asked about the frequency of food intake in the last month, which may mean that somewhat inaccurate information was collected. Third, the assessment of depression and anxiety symptoms through questionnaires is different from clinical diagnosis. Fourth, this study only performed a high-throughput sequencing of the 16S rDNA V3 + V4 region without metagenome; therefore, we could not analyze the further mechanisms of depression, anxiety, and gut microbiota.

## 5. Conclusions

In this study, depression and anxiety were found to be associated with constipation symptoms, constipation-related quality of life, and quality of sleep among older adults with FC. The worse the symptoms of constipation were, the higher the scores of depression and anxiety. The class Actinobacteria were negatively associated with depression and anxiety. Both mental disorders and abdominal symptoms were negatively associated with *g_Aquicella. G_Limnohabitans* was negatively associated with both mood disorders and rectal symptoms. Our study suggests that elderly people with FC generally have mood disorders: with the development of constipation symptoms, depressive and anxious moods increased. In the future, well-designed longitudinal studies to explore and confirm the role of the gut microbe in the relationship between constipation and emotional disorders are urgently required.

## Figures and Tables

**Figure 1 nutrients-14-05013-f001:**
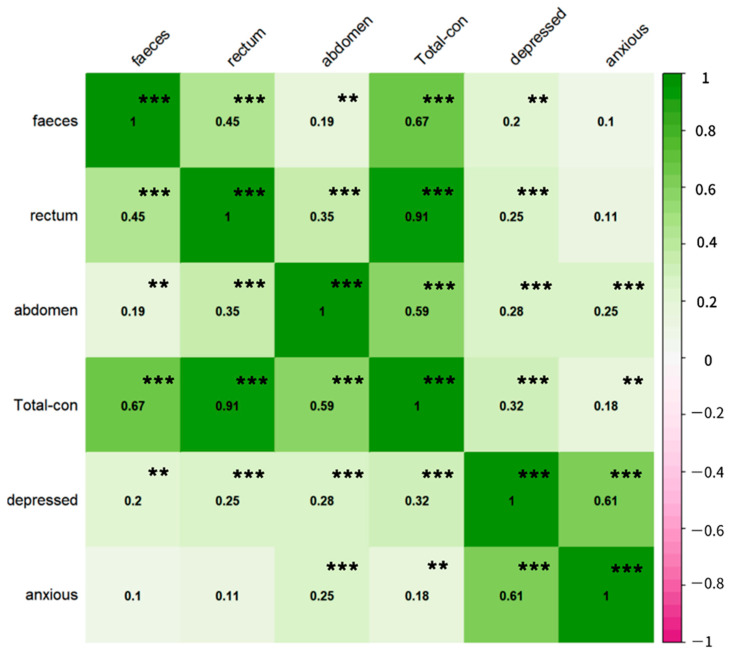
Correlation between depression, anxiety, and constipation symptoms. Faeces, rectum, and abdomen represent the dimensions of stool traits, rectal symptoms, and abdominal symptoms of PAC-SYM, respectively. Total-con: PAC-SYM (Patient Assessment of Constipation symptom) total score; depressed: PHQ-9 total score; anxious: GAD-7 total score. In addition, the correlation coefficient between anxiety and depression is high, 0.61 (*p* < 0.05), but through collinearity analysis, VIF = 1 of the two, and there is no evidence to show that there is collinearity. ** *p* < 0.01, *** *p* < 0.001.

**Figure 2 nutrients-14-05013-f002:**
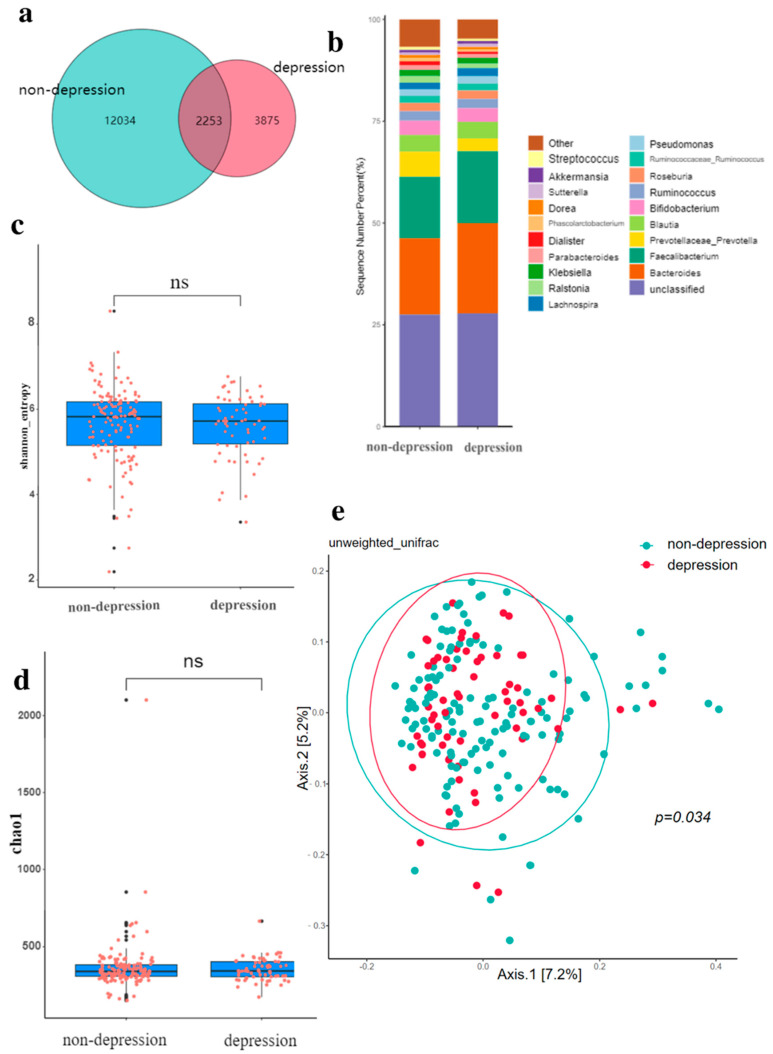
Gut microbiota in depression. (**a**) Comparison of intestinal microbiota OTUs between depression group and non-depression group. (**b**) The relative abundance of intestinal microbiota in the depressed group and the non-depressed group. (**c**) Shannon diversity of gut microbiota in depressed and non-depressed groups. (**d**) Chao-1 index of gut microbiota in depressed and non-depressed group. (**e**) Principal coordinate analysis (PCoA) of gut microbiota in depressed and non-depressed groups, *p* values were obtained by PERMANOVA and ANOSIM analysis. The black dots in (**c**,**d**) are outliers (values 1.5 standard deviations above the upper and lower edges of the box diagram). ns: no significance, *p* > 0.05.

**Figure 3 nutrients-14-05013-f003:**
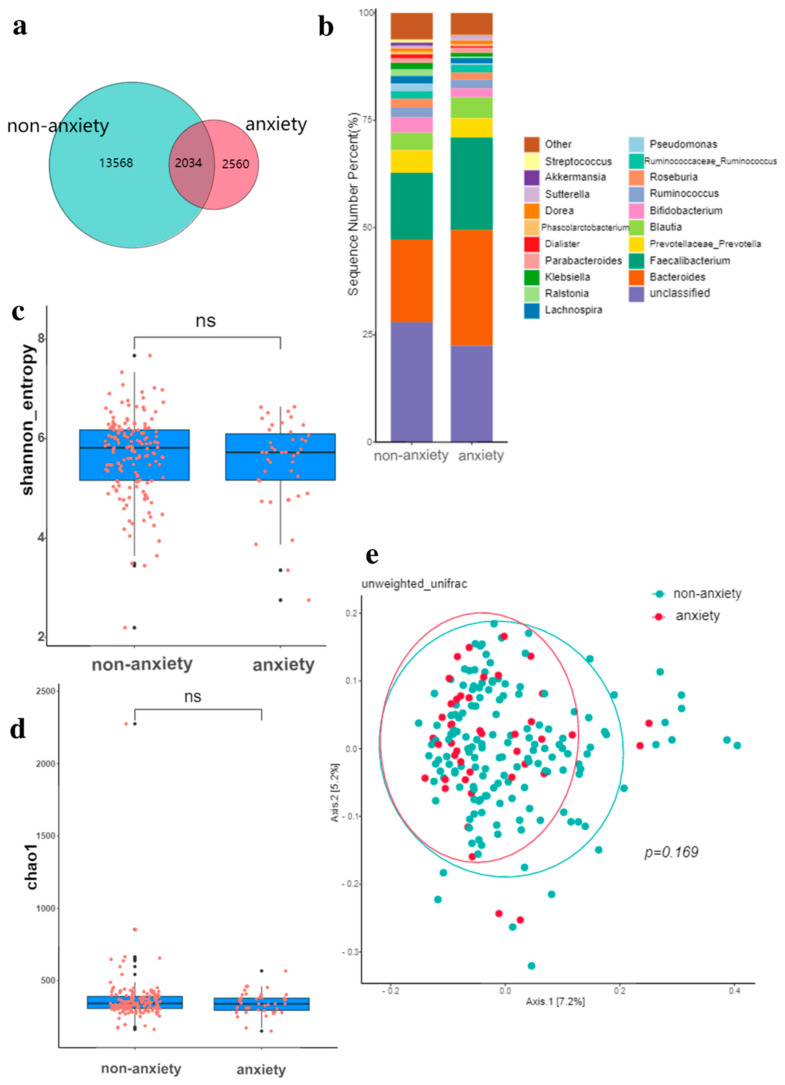
Gut microbiota in anxiety. (**a**) Comparison of intestinal microbiota OTUs between anxiety group and non-anxiety group. (**b**) The relative abundance of intestinal microbiota in the anxiety group and non-anxiety group. (**c**) Shannon diversity of gut microbiota in anxiety and non-anxiety groups. (**d**) Chao-1 index of gut microbiota in anxiety and non-anxiety groups. (**e**) Principal coordinate analysis (PCoA) of gut microbiota in anxiety and non-anxiety groups, *p* values were obtained by PERMANOVA and ANOSIM analysis. The black dots in (**c**,**d**) are outliers (values 1.5 standard deviations above the upper and lower edges of the box diagram). ns: no significance, *p* > 0.05.

**Figure 4 nutrients-14-05013-f004:**
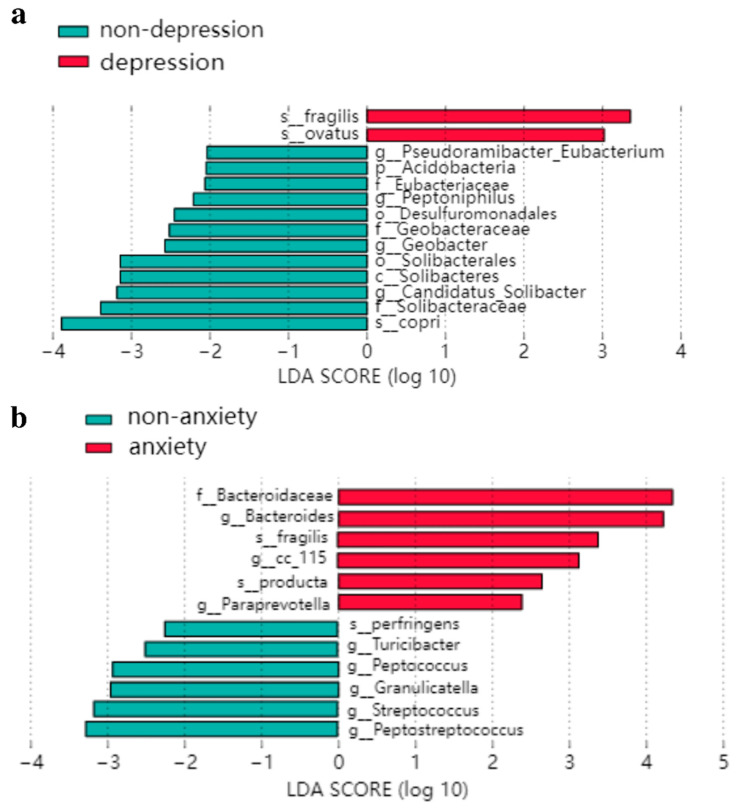
LEfSe analysis of the LDA histogram: (**a**) intestinal flora difference between depression group and non-depression group; (**b**) intestinal flora difference between anxiety group and non-anxiety group.

**Figure 5 nutrients-14-05013-f005:**
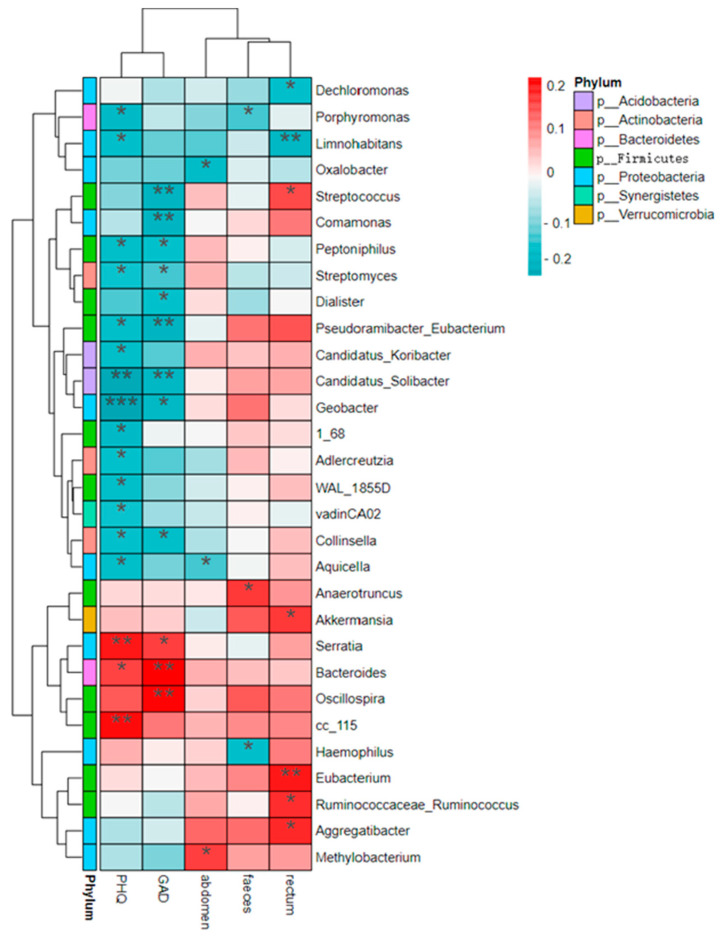
Heat map of interrelationships between genus level microbial species and phenotypes. Rectum, faeces, and abdomen represent the dimensions of rectal symptoms, stool traits, and abdominal symptoms of PAC-SYM, respectively. GAD and PHQ are the scores of GAD-7 and PHQ-9, respectively. * 0.01 ≤ *p* < 0.05, ** 0.001 ≤ *p* < 0.01, *** *p* < 0.001.

**Table 1 nutrients-14-05013-t001:** Basic characteristics of the research subjects (*n* = 198, *n* (%)).

Variables	Total	PAC-SYM			χ^2^	*p*
		Light 57 (28.8)	Moderate 78 (39.4)	Severe 63 (31.8)		
Age					6.491	0.165
60–69	81	18 (31.6)	32 (41.0)	31 (49.2)		
70–79	86	32 (56.1)	33 (42.3)	21 (33.3)		
≥80	31	7 (12.3)	13 (16.7)	11 (17.5)		
Sex					0.283	0.868
Male	85	26 (45.6)	32 (41.0)	27 (42.9)		
Female	113	31 (54.4)	46 (59.0)	36 (57.1)		
BMI (kg/m^2^)					11.572	0.021
<20	19	2 (3.5)	10 (12.8)	7 (11.1)		
20.0–26.9	140	36 (63.2)	58 (74.4)	46 (73.0)		
<26.9	39	19 (33.3)	10 (12.8)	10 (15.9)		
Residence *					0.046	0.977
Urban	191	55 (96.5)	75 (96.2)	61 (96.8)		
Town	7	2 (3.5)	3 (3.8)	2 (3.2)		
Education					6.434	0.376
Primary school or below	32	4 (7.0)	15 (19.2)	13 (20.6)		
middle school	69	23 (40.4)	28 (35.9)	18 (28.6)		
high school	78	24 (42.1)	27 (34.6)	27 (42.9)		
College or above	19	6 (10.5)	8 (10.3)	5 (7.9)		
Marital status *					3.040	0.551
Married	151	47 (82.5)	55 (70.5)	49 (77.8)		
Divorce	9	2 (3.5)	5 (6.4)	2 (3.2)		
Widowed	38	8 (14.0)	18 (23.1)	12 (19.0)		
Occupation before Retirement *					2.278	0.320
employed	182	55 (96.5)	70 (89.7)	8 (90.5)		
Other/unemployed	16	2 (3.5)	8 (10.3)	6 (9.5)		
Monthly household Income (RMB, yuan)					3.351	0.764
<2000	52	13 (22.8)	23 (29.5)	16 (25.4)		
2000–5999	81	23 (40.4)	29 (37.2)	29 (46.0)		
6000–10,000	50	17 (29.8)	18 (23.1)	15 (23.8)		
>10,000	15	4 (7.0)	8 (10.3)	3 (4.8)		
High dietary fiber intake					15.365	**0.004**
Low	68	14 (24.6)	23 (29.5)	31 (49.2)		
Moderate	65	20 (35.1)	34 (43.6)	11 (17.5)		
High	65	23 (40.4)	21 (26.9)	21 (33.3)		
IPAQ					1.545	0.819
Low	48	12 (21.1)	22 (28.2)	14 (22.2)		
Moderate	119	35 (61.4)	46 (59.0)	38 (60.3)		
High	31	10 (17.5)	10 (12.8)	11 (17.5)		

The Patient Assessment of Symptoms, PAC-SYM. The PAC-SYM scores were classified according to three-quarter cuts and compared by Chi-squared test. * Fisher’s exact test. Bold values indicate statistically significant values, *p* < 0.05.

**Table 2 nutrients-14-05013-t002:** Relationship between depression score, anxiety score, and constipation symptoms (*n* = 198, *n* (%)).

	Total	PHQ-9 ≥ 5	χ^2^	*p*	GAD-7 ≥ 5	χ^2^	*p*
		Mild 61 (30.8)			Mild 43 (21.7)		
PAC-SYM			11.802	**0.003**		10.619	**0.005**
Light	57	10 (17.5)			5 (8.8)		
Moderate	78	22 (28.2)			17 (21.8)		
Severe	63	29 (46.0)			21 (33.3)		
PAC-QoL			33.285	**<0.001**		30.723	**<0.001**
Low	66	36 (54.5)			29 (43.9)		
Moderate	63	19 (30.2)			10 (15.9)		
High	69	6 (8.7)			4 (5.8)		

Patient Health Questionnaire-9, PHQ-9. PHQ-9 ≥ 5 indicates at least mild depression. Generalized Anxiety Disorder-7, GAD-7. GAD-7 ≥ 5 indicates at least mild anxiety. The Patient Assessment of Symptoms, PAC-SYM. The Patient Assessment of Quality of Life, PAC-QoL. Comparisons were made using Chi-squared test. Bold values indicate statistically significant values, *p* < 0.05.

## Data Availability

The data that support the findings of this study are not publicly available, due to the data containing information that could compromise the participants’ privacy, but are available from the corresponding author upon reasonable request.

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
