# Peer review of "Association between Depression, Anxiety Symptoms and Gut Microbiota in Chinese Elderly with Functional Constipation"

_nutrients, 2022, doi:10.3390/nu14235013_

Round 1

Reviewer 1 Report

This is a very interesting study aiming to increase the knowledge on the role of the gut microbiota in the relationship between functional constipation and related mental disorders (anxiety and depression).

Minor revisions:

·      Use italics for all bacterial species: e.g. lines 275-277; 289-292.

·      Use capital letter for bacterial species: line 275 (s_ovatus); 291 (s_perfringens)

·      Correct: g_Pseudoramibacter-enbacterium to g_Pseudoramibacter-eubacterium (lines 355, 362); g_Aquilla to g_Aquicella (lines 327, 421).

·      Eliminate the contracted form, as it is not used further in the text: line 340 (CRF).

·      Use the extended form of the acronym: line 382 (SCFA).

·      Line 316: Does FC stand for abdominal and rectal symptoms? Please clarify the statement.

·      Use the extended form of the acronym in the legend: table 1 (PAC-SYM); table 2 (PHQ-9, GAD-7, PAC-SYM, PAC-QoL).

·      The title of Table 2 refers only to the “anxiety score”. Please also mention PHQ-9.

Major revisions:

·      There are several grammatical errors, please correct them: e.g. lines 69-70; line 91; line 123; line 143; line 271; lines 290-291; line 361; lines 363-364; line 404; lines 420-421.

·      Please describe in the Materials and Methods the LEfSe analysis.

·      A contradiction exists among these two statements: “The Class Actinobacteria were negatively associated with depression and anxiety…” (lines 316-317) and “The abundance of Actinobacteria increased … among individuals with mental disorders” (lines 326-328). Is there a positive or negative association between Actinobacteria and mental disorders? Please clarify.

·      Please make explicit that Actinobacteria, g_Aquicella and g_Limnohabitans are negatively or positively associated with emotional disorders and CF: lines 421-422.

·      Figures 2, 3, 4, and 5 present numerous details; however, they appear to be difficult to understand due to the low resolution. Please improve the graphics.

·      Figures should follow the text they are referring to. I suggest inserting Figure 2 after line 271 and Figure 3 after line 286. In addition, I recommend treating the LEfSe analysis of the two groups (anxious and depressed) in one paragraph and include Figure 4 below.

In the introduction and discussion the authors could profitably use the paper  by Gargari, Taverniti V et al. about Fecal Clostridiales distribution and short chain fatty acids reflect bowel habits…..Environ Microbiol. 2018; 20:3201-3213.

Reviewer 2 Report

This manuscript described the possible role of microbiota on the development of depression/anxiety in FC subjects, through the study among the elderly Chinese subjects.

Minor points to be added:

1.     Rationale for the selection of the elderly: Association between FC and depression/anxiety, low QOL or sleep problem is not specific to the elderly. Reason for the selection for the elderly should be included in the manuscript.

2.     Definition of the elderly: Subjects recruited for this study are age over 60 years old. This should be commented in the abstract, too.

3.     Wording for “Elderly”: The author used “older” (line 84, and twelve more), “senior” (line 103) and “elderly” (title and 23 more). This should be simplified. In the section of prevalence of depression and anxiety, those among “normal” elderly subjects in China cited as reference 31, which has different age groups for meta-analysis, such as over 55, over 60, over 65 and over 70. Dominant age group seems to be >60, which is similar to subjects in this manuscript. This can be mentioned.

4.     Defining depression and anxiety: The authors used PHQ-9 and GAD-7 for diagnosing depression and anxiety with the score 5 or more as depression/anxiety, respectively. In the description for defining depression and anxiety, the authors cited reference 20 for depression, reference 21 for anxiety. However, both references 20 and 21 described score 5 or more both for depression and anxiety as pathological. Therefore, both papers should be cited both for depression and anxiety.

5.     Wording in Table 2: In table 2, PHQ-9 Mild, GAD-7 Mild used as a group index. I don’t understand what “Mild” means. These seem to be PHQ-9 score >5, GAD-7 score >5, respectively, or PHQ-9 depression and GAD-7 anxiety.

6.     Sleep quality: The authors measured sleep quality using PSQI, and mentioned that impaired sleep quality in constipated patient, but the data is not shown. It should be shown somewhere in the manuscript.

Round 2

Reviewer 1 Report

The authors fully answered the questions/comments I had raised.

In my opinion the paper can be published